# Impact of Selected Sociodemographic and Clinical Parameters on Anxiety and Depression Symptoms in Paramedics in the Era of the COVID-19 Pandemic

**DOI:** 10.3390/ijerph19084478

**Published:** 2022-04-08

**Authors:** Kamil Piotr Marczewski, Magdalena Piegza, Alicja Zofia Gospodarczyk, Natalia Justyna Gospodarczyk, Krystyn Sosada

**Affiliations:** 1Department of Emergency Medicine, Faculty of Medical Sciences in Katowice, Medical University of Silesia, 40-635 Katowice, Poland; gospodarczyk.alicja@gmail.com (A.Z.G.); gospodarczykn02@gmail.com (N.J.G.); krystynsosada@wp.pl (K.S.); 2Department of Psychiatry, Faculty of Medical Sciences in Zabrze, Medical University of Silesia, 40-635 Katowice, Poland; mpiegza@sum.edu.pl

**Keywords:** COVID-19, pandemic, paramedics, anxiety, depression, sociodemographic factors

## Abstract

Introduction: The emergence of the SARS-CoV-2 virus, which caused the outbreak of the pandemic declared by the World Health Organization (WHO, World Health Organization) on 11 March 2020, has resulted in the introduction of many restrictions worldwide to contain the rapidly spreading pathogen. A particularly vulnerable professional group are paramedics working in Emergency Medical Teams. Aim: The main aim of this study was to investigate the influence of selected sociodemographic and clinical parameters on anxiety and depression symptoms in paramedics during the COVID-19 pandemic. Materials and methods: The study involved 387 paramedics working in Medical Rescue Teams in Poland. The majority of respondents were male (72.35%). In order to achieve the aim of the study, an online diagnostic survey was conducted using a questionnaire of the author’s own design and standardized questionnaires: Hospital Anxiety and Depression Scale (HADS) and General Anxiety Disorders (GAD-7). Results: Significantly higher values were observed for all analysed scales in females compared to males. The main factors influencing the occurrence of anxiety and depression symptoms were gender, seniority at work, family relations, use of sleeping pills. Alcohol consumption increased in this professional group during the pandemic. Conclusions: Females and users of sleep medication who work in the emergency department during a pandemic are more likely to experience symptoms of depression and anxiety. A longer length of service and satisfying relationships with family are factors in reducing these symptoms. Paramedics who are in informal relationships and single manifest more emotional problems compared to those who are married. Increased alcohol consumption can be considered as a way of coping with stress. However, further studies in this professional group are needed to assess the further impact of the pandemic on psychiatric symptoms among emergency medical workers.

## 1. Introduction

On 17 November 2019, the SARS-CoV-2 virus was first reported in the Chinese city of Wuhan, triggering a pandemic outbreak declared by the World Health Organization (WHO) on 11 March 2020. This has prompted a number of restrictions around the world to contain the rapidly spreading pathogen. WHO data show that, as of 21 February 2022, there had been 424 million cases worldwide, including 5.89 million deaths [1,2]. With increasing numbers of cases and deaths and inadequate health care systems, the stress of infection, death and loss of work has become a global problem. The media play an important role in spreading up-to-date information about the pandemic, providing daily data on new infections, hospital admissions and deaths. The fear caused by unpredictable health consequences and the need for isolation, often also from loved ones, have contributed to a high burden on psychological well-being [3]. People in quarantine have been diagnosed with emotional disturbances, including depressive symptoms, insomnia and symptoms of post-traumatic stress disorder (PTSD), and a number of these have been observed to make suicide attempts. People in at-risk groups, including pregnant females and those with young children, have been particularly concerned about the possibility of infection. Additional stressors included the disruption of daily routines, frustration, boredom, isolation from the world and an inability to meet basic needs, including going shopping. The stigmatisation of quarantined patients by their immediate environment was of great importance [4]. In a study by Vigil et al., paramedics were shown to be a particularly vulnerable group to stress, with their work being associated with an increased risk of depression, anxiety, sleep disorders, PTSD and suicide. During the COVID-19 pandemic, health care workers were even more vulnerable to mental health problems due to the severe stress of regular exposure to death and human suffering. In addition, fear of self-infection and inadequate personal protective equipment led to the abandonment of existing jobs. This loss of workforce increased the burden on remaining workers, requiring them to work longer hours and a heavier workload. The unpredictable nature of the new virus and the fear for their own health and that of their family undoubtedly has had a negative impact on the psychological state of paramedics [5].The authors of the present study attempted to determine the relationship between selected sociodemographic and clinical parameters and the level of anxiety and intensity of depressive symptoms in a group of paramedics.

## 2. Materials and Methods

### 2.1. Procedure and Participants

In order to achieve the aim of the study, an online diagnostic survey was used, using the technique of self-administered questionnaires and standardized questionnaires: Hospital Anxiety and Depression Scale (HADS) and General Anxiety Disorders (GAD-7). They were made available from 10 December 2020–14 January 2021 on social networking sites (Facebook—professional groups for paramedics, Instagram) and online forums dedicated to paramedics. A total of 387 paramedics working in Medical Rescue Teams across Poland took part in the study—280 male (72.35%) and 107 female (27.65%). Sociodemographic characteristics of the study group are presented in Table 1. Inclusion criteria were: age over 21 years, occupation as a paramedic, active during the COVID-19 pandemic, informed consent to participate in the study. Exclusion criteria were: not consenting to participate in the study, impaired consciousness, not professionally active during the COVID-19 pandemic, retired.

The HADS scale was introduced by Zigmond and Snaith in 1983 and belongs to self-assessment scales [6]. The Polish version was developed by M. Majkowicz, K. de Walden-Gałuszko and G. Chojnacka-Szawłowska. The questionnaire consists of two independent subscales to assess anxiety and depression symptoms, and each subscale has seven statements. The questionnaire was augmented by two statements on the level of aggression. Answers are given on a 4-point Likert scale (0–3). The final score for each subscale is between 0–21 points. The score for the two questions concerning aggression ranges from 0–6 points. Normal scores range from 0–7, a range of 8–10 points indicates borderline levels, while values of 11–21 are considered abnormal. In both anxiety and depression subscales, scores are summed. Values from 0 to 7 indicate normal, 8 to 10 are borderline (discrete anxiety symptoms) and values from 11 to 14 are considered moderate anxiety symptoms, 15–21 points are considered severe anxiety symptoms (anxiety syndrome) [6,7]. The GAD-7 scale is a 7-item questionnaire designed to assess the severity of anxiety symptoms in the past 2 weeks, which, however, is not the same as a medical diagnosis of generalised anxiety disorder. The questions in the questionnaire address the extent to which the patient has been bothered by feelings of nervousness, being on edge, inability to stop or control worrying, problems relaxing and other markers of anxiety. The questionnaire was developed by Spitzer et al. [6] and published in 2006. It is quick and easy to use and is currently being used in research and clinical settings. The copyright is owned by Pfizer Inc., although use of the questionnaire is free. The questionnaire has been validated for use as a screening tool [8,9] in primary care [10] and in the general population [11].

Using a threshold score of 10, the GAD-7 has a sensitivity of 89% and specificity of 82% for generalised anxiety disorder. It is also moderately good for screening targeting other anxiety disorders—paroxysmal anxiety syndrome (sensitivity 74%, specificity 81%), social phobia (sensitivity 72%, specificity 80%) and post-traumatic stress disorder (sensitivity 66%, specificity 81%) [11]. A score of 5, 10, 15 corresponds to mild, moderate and severe anxiety, respectively. A sum of at least 10 points indicates a high probability of generalized anxiety syndrome [12]. Cronbach’s alpha is the most commonly used reliability coefficient for questionnaires in psychology, demonstrating the internal consistency of the tool. For the AIS (Athens insomnia scale) alpha this value was 0.866, the Epworth scale alpha = 0.8354, the HADS questionnaire alpha = 0.807 and the GAD7 questionnaire alpha = 0.9164.

The study used a self-administered questionnaire, which included questions about the analysis of the following factors: gender, marital status, assessment of relationships with family members during COVID-19 pandemic, alcohol consumption before and during the pandemic, taking sleeping pills and psychoactive drugs, and length of service in Emergency Medical Teams. The assessment of family relationships during the pandemic was presented in the form of an 11-point scale (from 0 to 10), where 5 meant the existence of family relationships at the same level as before the pandemic, from 6–10 the improvement of family ties, while from 4–0 the deterioration of relationships with family members, compared to the period before the pandemic. The sub-question on the use of psychoactive substances during the COVID-19 pandemic included the use of marijuana, cocaine, amphetamine, heroin, mephedrone. In the question on the use of sleeping pills, no specific medicinal substances were specified.

### 2.2. Data Analysis

Excel 2021 and the R language in the RStudio environment, using the pROC package, were used to analyse the data collected. Data were presented as the number of people indicating a given answer and the percentage of the total population. Quantitative variables were presented as mean +/− standard deviation. Alcohol consumption was ordered on an eight-point scale, where 1 meant no consumption and 8 meant daily alcohol consumption, and was further analysed as a quantitative variable. For comparisons of quantitative variables, the Student’s *t*-test with Forsyth Brown’s test for homogeneity of variance was used. For comparisons of more than two samples, ANOVA analysis of variance was used, with Tukey’s post hoc test for unequal counts. The model was performed using the backward stepwise method. Values of *p* < 0.05 were considered statistically significant.

### 2.3. Ethics

The study was performed with the approval of the University Bioethics Committee of the Medical University of Silesia in Katowice (PCN/CBN/022/KB/233/121) and was in accordance with the Declaration of Helsinki. Each participant, before taking part in the study, was informed about the anonymity and full voluntary participation in the study with the possibility of stopping participation in the survey at any time.

## 3. Results

The study involved 387 paramedics working in Medical Rescue Teams in Poland. The majority of respondents were male (72.35%). A detailed description of the study group is presented in Table 1.

Table 2 shows the general characteristics of the group. Both females and males were included in the statistical analyses. The mean number of points obtained in the HADS-A subscale is 8.88, the minimum score is 2.00 and the maximum score is 18.00 points.

The mean score of the HADS-D subscale is 6.35, the minimum score is 0.00 and the maximum score is 17.00 points. The mean number of points obtained on the GAD-7 scale is 8.28, the minimum score is 0.00 and the maximum score is 21.00 points.

Figure 1 shows a comparison of the HADS and GAD-7 scale values by gender. Significantly higher values of all analysed scales were observed in females as compared to males. In the HADS-A scale, 25.2% of females achieved a score of 0–7, which is considered normal, 23.4% achieved a total score qualifying for discrete disorders, 39.3% achieved scores indicating moderate disorders and 12.1% showed severe disorders. On the HADS-D scale, 47.7% of the females achieved a score of 0–7, which is considered normal, 28.00% achieved a sum score qualifying for discrete disorders, 21.5% achieved a score indicating moderate disorders and 2.8% showed severe disorders. On the GAD-7 scale, 16.8% of the females achieved a score of 0–5, which qualifies for mild anxiety disorder, 35.5% achieved a total qualifying for moderate anxiety disorder and 47.7% achieved a score indicating severe anxiety disorder.

Table 3 shows the comparison of scale values by marital status of the subjects. HADS-D scores were significantly higher in single compared to married persons, with no significant difference to unmarried persons. Scores on the GAD-7 scale differed significantly between married participants and those in informal relationships or single individuals. In total, 40.7% of married people, 30.9% of single people and 28.3% of those in informal couples scored 0–5, indicating a mild anxiety disorder, while 32.5% of married persons, 33.0% of single persons and 35.9% of persons in an informal relationship scored 6–10, indicating a moderate anxiety disorder. Finally, 26.8% of married persons, 36.2% of single persons and 35.9% of persons in an informal relationship scored 11–15, indicating severe anxiety disorder.

Figure 2 shows the relationship between the scores on the scales used and the use of sleep aids. Slightly higher scores were found in respondents using sleep medication. During the COVID-19 pandemic, up to 20.3% of paramedics started using sleeping pills.

Table 4 illustrates the relationship between the use of psychoactive drugs (excluding sleeping pills) by the respondents and the scale scores. A borderline relationship was observed. In the survey conducted, as many as 7.7% of the respondents admitted having used psychoactive substances (marijuana, cocaine, amphetamine, heroin, mephedrone) while in quarantine/isolation.

Table 5 shows the correlation between the scale scores and the assessment of the quality of the relationship with the family. A strongly negative correlation was shown. This indicated that the worse the relationship with family members, the higher the respondents scored in all scales. In total, 34.4% of the respondents indicated a deterioration in family relationships during the COVID-19 pandemic, while 28.3% of the emergency workers indicated that their relationships had not changed.

Table 6 compares the correlations between age and length of service. Significant negative correlations were observed between all scales and age and seniority. Seniority has a greater influence on the correlation than the age of the subject alone. Persons with longer work experience achieved lower scores in all scales used.

Table 7 shows the percentage increase in alcohol consumption among paramedics working in Emergency Medical Teams. In total, 17.2% of respondents reported drinking 2–3 times a week before the pandemic began. During the study period, the percentage of those consuming alcohol 2–3 times a week increased to 30.6%.

## 4. Discussion

Studies conducted during the COVID-19 pandemic clearly show its significant impact on mental health among health care workers. The aim of this study was to assess the impact of sociodemographic factors and selected clinical parameters on depressive and anxiety symptoms among paramedics working in Emergency Medical Teams (EMTs) in Poland. Emergency responders have a great responsibility for the health and lives of others and their own, which is an important aspect of public health in times of pandemics. At the time of writing, the authors have not reported a similar study describing the mental state of paramedics working in the Emergency Medical Service during the COVID-19 pandemic. According to currently available knowledge, this is the first study addressing this topic.

During the period of our project, Poland was struggling with the second wave of COVID-19. The Ministry of Health reported 27,875 new coronavirus infections in Poland, with an average of 349 deaths per day, as of 7 November 2020 [13]. Among 387 employees of Emergency Medical Teams, as many as 25.4% of males and 47.7% of females scored 11–15 points on the GAD-7 scale, which indicates severe generalized anxiety. Such a high percentage of positive results may be caused by insufficient protection in personal protective equipment, staff shortages, lack of adequate knowledge about the SARS-CoV-2 virus, social limitations (lack of opportunities for family meetings, going out to restaurants with friends, holiday trips, etc.) and the need to stay in isolation/quarantine away from the immediate family, as well as helplessness and powerlessness in the fight against the pandemic. Limiting to a large extent the possibility of direct interpersonal contacts, according to specialists, has a huge impact on maintaining proper hygiene of human mental health [14,15]. The results obtained in the present study confirm the works of many authors. According to Sagar et al. in their meta-analysis conducted in India, the main risk factors for anxiety are female gender, young age group with low work experience and marital status. The prevalence of anxiety symptoms was higher in females (21.7%) than in males (16.2%) and the level of anxiety intensity was higher in the young age group and in unmarried individuals [15]. According to the authors of the cited publication, this may result from the fact that younger health care workers have less experience and fewer developed methods of coping with stress at work. The authors of a study conducted in Bangladesh presented that females are more likely to experience symptoms of anxiety and depression, which is consistent with our results [16,17]. In Poland, paramedics with shorter work experience are mainly deployed in teams to patients infected and suspected of being infected with SARS-CoV-2. Similar conclusions were reached by Chinese researchers who, during the COVID-19 pandemic, conducted a cross-sectional survey on a group of 7236 people, 2250 of whom were health care workers. The prevalence of anxiety disorders in them was 35.6%, and depressive symptoms were reported by 19.8% of respondents, while poor sleep quality was demonstrated in 23.6% of people. Depression, generalized anxiety disorder and poorer sleep quality were more common in those younger than 35 years [18]. In Jiangsu Province, China, 1521 health care workers were surveyed during the COVID-19 pandemic. The prevalence of mental disorders in the study group was 14.1%, of which this prevalence was 12.3% in males and 14.7% in females (depressive symptoms, anxiety). Mental disorders were found less frequently in those aged over 41 years. In contrast, health care workers with less experience in life and health-threatening situations showed more frequent mental abnormalities. These abnormalities were more common among medical than non-medical staff, with medical staff having the highest percentage. It was found that a lower risk of mental disorders was present in married workers. Having offspring was also a protective factor; interestingly, having two or more children was more protective than having only one [19]. Moreover, in our study, health care professionals with more than 20 years of work experience reported fewer anxiety and depressive symptoms. This is probably the result of their clinical knowledge and the experience and maturity gained over the years [20]. Another study also confirmed that a higher level of education (bachelor’s or master’s degree) and work experience of more than 20 years are protective factors against the occurrence of anxiety and depressive symptoms [21]. A study by Guo et al. who assessed the impact of the COVID-19 pandemic on emotional disturbances among 11,118 Chinese hospital workers found that 18–57% of medical staff experienced them during the pandemic period [22]. The adverse impact of the COVID-19 pandemic was also confirmed in another study—among 1257 Chinese covid workers, more than half experienced depressive symptoms and 45% reported anxiety disorders [23]. Other researchers have also come to similar conclusions. A study by Huang et al. on the prevalence of generalised anxiety and depressive symptoms in a group of 7236 Chinese healthcare workers, reported the significant impact of the COVID-19 pandemic on the mental health of study participants [18].

In another cross-sectional study, Songül Araç et al., based on HADS scale scores, showed that a large proportion of participants scored 11 and above, indicating the presence of severe anxiety and depressive symptoms. In addition, 39.4% of participants scored above 11 on the HADS-A, indicating anxiety symptoms of severe severity, and 31.3% of study participants scored above 11 on the HADS-D, indicating depressive symptoms of moderate to severe severity [24]. In our study, 33.08% of paramedics presented severe severity of anxiety symptoms, scoring above 11 points in the HADS-A questionnaire. A difference appeared in the score obtained in the HADS-D scale, where only 13.95% of the respondents obtained a score above 11 points, indicating the presence of depressive symptoms of severe intensity. This may be due to the higher proportion of females participating in the experiment of Songül Araç et al. where females accounted for 37.5% of the subjects.

The observation by Wong et al. shows that stress levels and severity of stress-related anxiety symptoms were significantly higher among emergency personnel, which was also confirmed in our study and is consistent with the literature data [25].

A study by Rafia Tasnim et al. confirmed the prevalence of anxiety and depressive symptoms among health care workers in Bangladesh during the COVID-19 pandemic. In their study, 69.5% of the subjects exhibited anxiety symptoms and 38.5% revealed depressive symptoms [26]. A cross-sectional study conducted in 34 hospitals in China presented similar findings: that health care workers in direct contact with SARS-CoV-2-infected patients experience mental health problems, including exhibiting symptoms of anxiety and depression. An earlier Chinese study provided estimates of the prevalence of anxiety symptoms (46% compared to 31.52% by our team using the Generalised Anxiety Scale (GAD-7)) among health care workers working during the COVID-19 pandemic [23]. Most researchers agree that the COVID-19 pandemic is a particular period when there has been a significant increase in the proportion of people employed in medical institutions with features of generalized anxiety syndrome. Taking into account the insufficient number of paramedics in Poland, the shortage of personal protective equipment (at the beginning of the COVID-19 pandemic) and the fear of infecting oneself and others with the SARS-CoV-2 virus, one should expect either a further increase in the percentage of people with emotional disorders or its decrease as a result of adaptation to the conditions of pandemic stress. However, this issue requires further research.

At present, there are no publications describing the impact of the pandemic on family relationships among health care professionals, including paramedics. Medical personnel chronically exposed to stress are more likely to experience increased psychological strain, which not only results in reduced psychological well-being but also negatively affects social and family relationships [27]. The increased risk of infection in the course of COVID-19 for the elderly has resulted in many families deciding to limit contact with grandparents and other family members who, before the pandemic, helped their children and relatives to raise the next generation, running entire households of their loved ones. During the pandemic period, parents also had to take on a significant part of the responsibilities of teachers and educators of their children. Health care personnel additionally had to cope with the challenge of the risks that their job entailed for their immediate environment. Some health care professionals decided to voluntarily isolate themselves from their families. In a survey conducted by our team, 34.6% of respondents indicated a deterioration in their family relationships, 28.3% responded that their relationships did not change during the pandemic and 37.00% of Emergency Medical Teams staff indicated an improvement in their relationships with family members.

To the best of our knowledge, there are currently no studies directly addressing the use of alcohol, psychoactive substances and sleeping pills among paramedics. This article is the first to describe this phenomenon during the COVID-19 pandemic in relation to this professional group. According to the study conducted by Paweł Rasmus et al., paramedics in Poland working in the Emergency Medical Service most commonly use alcohol (32.6%); much less frequently psychoactive drugs (10.9%) and sleeping pills (2.2%) [28]. It is worth noting that these data refer to the period before the COVID-19 pandemic. It was shown that alcohol consumption increased in 30.6% of people during the pandemic compared to the period before, in which 17.2% of respondents reported drinking alcohol 2–3 times a week. In our study, 20.3% of respondents admitted to using sleeping pills during the pandemic and 7.7% of paramedics admitted to using psychoactive substances while in quarantine or isolation.

## 5. Limitations of the Study

The online survey we conducted has some limitations. In order to realise the research assumption, apart from the structured mental health assessment scales, we used the author’s questionnaire, which was not validated. The self-assessment questionnaires used do not provide sufficient grounds for making a psychiatric diagnosis, they can only inform about the severity of symptoms and suggest specific disorders. Due to the lack of direct contact, it is impossible to be 100% sure that the person invited to the survey is really who he/she claims to be, which reduces the reliability of the opinions obtained. On the other hand, internet surveys are a recognised method of obtaining data that are considered reliable, if only for the reason that they eliminate the stress associated with direct contact with the person being surveyed.

## 6. Conclusions

Females and users (both female and male) of sleep medication working in Emergency Medical Teams scored significantly higher in all analysed scales assessing emotional state.Those with more years of work and declaring better relations with their families achieved lower values in all scales.Those in informal relationships and those who were single showed more intense emotional problems compared to those in married relationships.Alcohol consumption increased in 30.6% of people during the pandemic compared to the period before, which can be considered a way of coping with stress.Male gender, marriage, reported better relationships with family, non-use of sleep medications and longer length of service appeared to be protective factors against emotional disturbance in the era of COVID-19 pandemic in paramedics.

The COVID-19 pandemic has caused severe psychological distress among paramedics working in the Emergency Department. Therefore, it is extremely important to develop effective stress management strategies that should be implemented as early as possible to prevent the development of chronic stress complications. Additionally, paramedics should be trained to adequately protect and care for their mental well-being. Mental health services should be easily accessible, and support programmes should be developed to reduce the risk of disorders occurring in this vulnerable group of health care workers. However, further research is needed to evaluate the effectiveness of stress coping methods to improve the mental state of paramedics during the COVID-19 pandemic.

## Figures and Tables

**Figure 1 ijerph-19-04478-f001:**
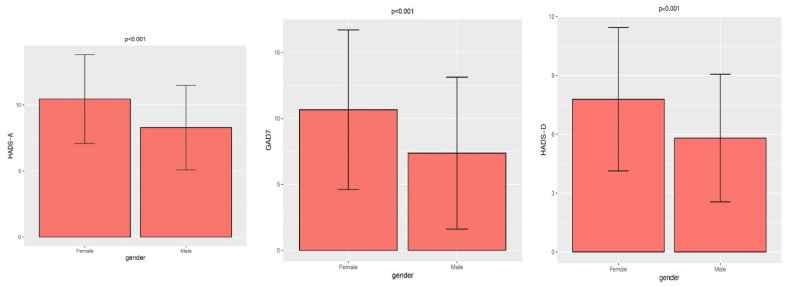
Comparison of HADS and GAD-17 scale values by gender using the Student *t*-test.

**Figure 2 ijerph-19-04478-f002:**
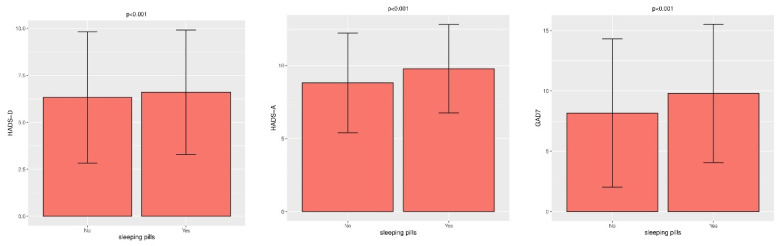
Relationship between the indications of the HADS and GAD-7 scales and the use of sleep aids using the Student *t*-test.

**Table 1 ijerph-19-04478-t001:** Sociodemographic characteristics of the study group.

	n	%
**Gender**		
**Male**	280	72.35
**Female**	107	27.65
**Size of the town where you work?**		
**city with up to 50,000 inhabitants**	123	31.78
**city with 50,000 to 100,000 inhabitants**	106	27.39
**city between 100,000 and 500,000 inhabitants**	81	20.93
**city with more than 500,000 inhabitants**	77	19.90
**Marital status**		
**married**	194	50.13
**free**	95	24.55
**casual relationship**	92	23.77
**widower**	3	0.78
**Divorced**	3	0.78
**Do you have children?**		
**Do not**	183	47.29
**Yes, at school age**	72	18.60
**Yes, up to 3 years**	47	12.14
**Yes, at pre-school age**	46	11.89
**Yes, during adolescence (12–16 years of age)**	39	10.08

**Table 2 ijerph-19-04478-t002:** General characteristics of the group in relation to the scales analysed.

Variable	No of Observations	Average	Minimum	Maximum	Standard Deviation
HADS-A	387	8.88	2.00	18.00	3.39
HADS-D	387	6.35	0.00	17.00	3.48
GAD7	387	8.28	0.00	21.00	5.75

**Table 3 ijerph-19-04478-t003:** Comparison of HADS and GAD-7 scale values by marital status.

Variable	Married		Unmarried		Casual Relationship		*p*
	Average	SD	Average	SD	Average	SD	
HADS-A	8.65	3.45	8.93	3.46	9.27	3.06	0.34
HADS-D	5.71	3.26	7.04	3.84	6.76	3.17	*0.00*
GAD7	7.39	5.60	8.93	5.64	9.42	5.85	*0.01*

**Table 4 ijerph-19-04478-t004:** Use of psychoactive drugs by respondents and results of the HADS and GAD-7 scales.

Variable	Did Not Use		Psychoactive		*p*
	Average	SD	Average	SD	
HADS-A	8.81	3.41	9.79	3.04	0.14
HADS-D	6.33	3.49	6.61	3.31	0.69
GAD7	8.17	5.72	9.79	5.95	0.15

**Table 5 ijerph-19-04478-t005:** Correlations between HADS and GAD-7 scales indications and relationship assessment with family using Pearson correlation coefficient, (results in bold at *p* < 0.05).

Variable	How Would You Rate Your Relationship with Your Family Members during the COVID-19 Pandemic?
HADS-A	−0.24
HADS-D	−0.30
GAD7	−0.30

**Table 6 ijerph-19-04478-t006:** Correlations between age and length of service using Pearson correlation coefficient, (results in bold at *p* < 0.05).

	Age	Length of Service in the Emergency Medical Service
HADS-A	−0.14	−0.18
HADS-D	−0.14	−0.17
GAD7	−0.21	−0.25

**Table 7 ijerph-19-04478-t007:** Comparison of alcohol consumption 2–3 times per week before and during the pandemic period.

Percentage of Alcohol Consumption before the Pandemic	Percentage of Alcohol Consumption during the Pandemic
17.2%	30.6%

## Data Availability

Data supporting reported results are available on request from the study team.

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
