# Peer review of "Impact of Selected Sociodemographic and Clinical Parameters on Anxiety and Depression Symptoms in Paramedics in the Era of the COVID-19 Pandemic"

_ijerph, 2022, doi:10.3390/ijerph19084478_

Round 1

Reviewer 1 Report

Thank you very much for give me the opportunity to review this interesting paper entittled Impact of selected sociodemographic and clinical parameters on anxiety and depression symptoms in paramedics in the era of the COVID-19 pandemic.

Altogether it is an interesting study aimed to to investigate the influence of selected 16 sociodemographic and clinical parameters on anxiety and depression symptoms in paramedics during the COVID-19 pandemic, with a propper methodology. In the authors' own words, according to currently available knowledge, this is the first study addressing this topic. Said this, I have some concerns.

  1. In my modest opinion, it would be convenient to help the reader a little more to follow the authors. For example, in your introduction I don´t find references to why in this specific population it is important to address how certain sociodemographic variables can be related to the symptoms of anxiety and depression, nor about which of all of them the research seems to indicate that are the most important in this context. However, you comment some of these issues in your discussion – that it is quite good- without giving the reader a clearer idea of the state of the question in the introduction (e.g. The results obtained in the present study confirm the works of many authors. According to Sagar et al. in their meta-analysis conducted in India, the main risk 218 factors for anxiety are female gender, young age group with low work experience and 219 marital status). My suggestion would be to reinforce that introductory section, this would also allow you to incorporate more bibliographical references of interest in your work.
  2. Your conclusions section lists a summary of the main findings obtained, it would be interesting if it also be offered suggestions regarding what strategies can be used to intervene on the detected problem. I would also suggest writing this section instead of using numbers, with a little more elaborate wording and incluiding a little more detail regarding practical implications of you study.
  3. Perhaps you can use the expression procedure and participants, instead of data and participants (line 112)
  4. I do not find in your study what is the reliability of the measures you use in your sample, for example indicating alpha values, could you incorporate them?
  5. Please check the format, for example, in line 83, and maybe you have to use italic for significant p, for example in line 122 (p<0.05)

Author Response

Thank you very much for your valuable suggestions and comments on the article. I tried to make all the changes you pointed out.

1) In the introduction, I have included information on why it is important to assess socio-demographic factors influencing depression and anxiety disorders in this population during the COVID-19 pandemic. In addition, I have referred to the results published in the journal indicated in the bibliography.

2) In the results section, I have included information on strategies that can be implemented to improve mental state conditioning among Emergency Medical Services Team members.

  1. the phrase "data and participants" was changed to "procedure and participants"
  2. I introduced information on the reliability of the measures as Cronbach's alpha coefficient.
  3. I used italics for significant p<0.05.

If you have any questions, please feel free to contact me. At the same time, I again ask for further evaluation and guidance.

Reviewer 2 Report

Dear authors! Thank you for the research, the scientific significance of which is difficult to overestimate.

However, I would like to draw your attention to some points that should be corrected.
1. The article should replace women into female and men into male
Summary in the conclusion section, you write "Conclusions: Women and users of sleep medication". In my opinion, in a conclusion related to gender, identification should be followed. Therefore, it is better to clarify the status of those whom you call "users"
2. The materials and methods do not sufficiently disclose the criteria by which the participants were selected.
3. In table 1, you need to uniformize the participants by indicating female and male
4. In figure 1, the designation of the column should be clarified. You have gender, and in the signature in one case there is the letter "M" (probably it is male), and in the other case it is "K".

Author Response

Dear reviewer,

Thank you very much for your valuable suggestions and comments on the article. I tried to make all the changes you pointed out.

1.The forms have been changed from women to female and men to male. We have also separated users by gender in the conclusions.

  1. I introduced inclusion and exclusion criteria in the materials and methods section of the study.
  2. We did not perform a gender split due to the lack of need for separation for statistical analysis. Making changes may result in an overly expanded table, which may reduce its readability.

4.I have introduced standardised gender designation in the figures. If you have any questions, please feel free to contact me. At the same time, I again ask for further evaluation and guidance.

If you have any questions, please feel free to contact me. At the same time, I again ask for further evaluation and guidance.

Round 2

Reviewer 1 Report

Thank you for giving me the opportunity to review the manuscript again. In my opinion the authors have made substantial improvements with a good result.